# Availability of Hib, Pneumococcal, Rotavirus, and HPV vaccines in China: Implication for equity in access to immunization services

Lei Guo[1]*, Feng Guo[2]*, Weixi Jiang[3], Xinyu Zhang[1], Di Dong[4], Shu Chen[5], Quan Wan[2], Shenglan Tang[1,5,6]

1 Global Health Research Center, Duke Kunshan University, Kunshan, China, 2 Department of Health Economics and Healthcare Security, China National Health Development Research Center, Beijing, China, 3 School of Public Health, Fudan University, Shanghai, China, 4 The World Bank Group, Washington D.C, United Stated of America, 5 Duke Global Health Institute, Duke University, Durham, North Carolina, United States of America, 6 SingHealth Duke-NUS Global Health Institute, Duke-NUS Medical School, Singapore, Singapore

* lei.guo@dukekunshan.edu.cn (LG); gf@nhei.cn (FG)

## Abstract

Vaccines are cornerstones of public health, yet equitable access remains a major global challenge. While China has achieved sustained high coverage of National Immunization Program (NIP) vaccines, some WHO-recommended cost-effective non-NIP vaccines, such as Hib, pneumococcal conjugate (PCV), rotavirus, and human papillomavirus vaccines (HPV) are not universally accessible and often require out-of-pocket payment. This study examines geographic and urban–rural disparities in the availability of these four vaccines across China using data from a nationally representative China's National Health Accounts survey in 2022, covering 355 district- and county-level disease control centers and 2,609 public primary health facilities across 27 provinces. Availability was defined as the presence of at least one dose delivered or administered. Inequality was assessed using Slope Index of Inequality (SII) and the Erreygers-corrected Concentration Index (CI), based on provincial GDP per capita. We found significant disparities in vaccine availability across economic development levels and urban–rural settings in China. Overall, HPV vaccines were most available (96.9%) and PCV13 the least (86.5%) at the district/ county level. Urban areas had markedly higher availability than rural areas, particularly for PCV13 (94.2% vs. 79.1%). At the clinic level, over 70% of urban clinics provided all four vaccine types, compared to just 40% in rural areas. The vaccines with the highest inequality were DTaP-IPV-Hib (CI: 0.408), PCV13 (CI: 0.362), and RV5 (CI: 0.343), indicating strong pro-rich distribution. In contrast, the Hib vaccine demonstrated the lowest inequality (CI: 0.209). These findings highlight significant inequities in the availability of WHO-recommended non-NIP vaccines across China, particularly affecting rural and economically disadvantaged regions underscoring the

---

**Data availability statement:** The data used in this study were collected through the 2022 China National Health Accounts Study coordinated by the China National Health Development Research Center (NHDRC). The dataset includes aggregated administrative records on vaccine procurement and usage at the district or county and clinic levels and is part of the National Health Accounts study. Due to legal and administrative restrictions under regulations issued by government departments, the data cannot be publicly shared. Researchers may request access to the data from the China National Health Development Research Center (NHDRC), subject to institutional review and approval. Other economic data used in this study are publicly available from the National Bureau of Statistics database (https://data.stats.gov.cn/).

**Funding:** The work reported in this publication is part of the research project Innovation Lab of Vaccine Delivery Research, supported by the Gates Foundation (INV034554 to ST). The funders had no role in study design, data collection and analysis, decision to publish, or preparation of the manuscript. The conclusions and opinions expressed in this work are those of the authors alone and shall not be attributed to the Foundation.

**Competing interests:** The authors have declared that no competing interests exist.

need of expanding vaccine provision in underserved areas, strengthening rural service capacity, and considering the inclusion of these vaccines in the NIP.

## Introduction

Vaccines are one of the most crucial and cost-effective tools in public health, playing a pivotal role in preventing infectious diseases and ensuring global health security. While significant strides have been made in vaccine development, the equally critical aspect of vaccine availability remains under-addressed, particularly in regions with limited resources [1,2]. Immunization Agenda 2030 underscores the importance of vaccine equity, urging all countries to enhance their efforts in this area [3]. However, equitable access to vaccines continues to pose a substantial challenge. A World Health Organization (WHO) report indicated that 68 countries reported at least one national stock-out event in 2023, with leading vaccines including DT-containing, OPV, measles-containing, BCG, Td-containing vaccines, with procurement and funding delays being the main reasons [4].

In China, attention to vaccine availability has been emphasized, but remains uneven across regions with different levels of socio-economic development. As highlighted by the Vaccine Administration Law of the People's Republic of China, vaccine products on the Chinese market fall into two categories: those included in the National Immunization Program (NIP), which are provided free of charge to children under six years old, and non-NIP vaccines, which require out-of-pocket payments [5,6]. NIP vaccines are funded by government budgets, ensuring universal availability regardless of economic status or geographic location of vaccine recipients. Their provision was largely mandated by the Law and implemented by governments at all levels. In contrast, non-NIP vaccines are often used less frequently due to higher prices, insufficient awareness, and vaccine hesitancy, as well as supply issues [5,7]. Among all vaccines recommended by the WHO to be included in the NIP, four of them, namely, Haemophilus influenzae type b (Hib), pneumococcal conjugate vaccine (PCV), rotavirus, and human papillomavirus (HPV) vaccines, have not yet been included in China's NIP [8,9]. In addition, the non-NIP vaccines also play a critical role in reducing the burden of vaccine-preventable diseases beyond childhood, particularly those associated with high disease burdens such as meningitis, pneumococcal infections, and diarrheal diseases in China [10].

Previous studies examined the degree of equity in access to and coverage of vaccines in China, mostly from a demand-side perspective, but there is limited research examining subnational disparities in terms of vaccine availability from the supply perspective [7,11]. In addition, less attention has been paid to equity in the availability of these WHO-recommended vaccines that are not yet included in China's NIP. This study aims to investigate regional and urban-rural disparities in vaccine availability across China. By examining the availability of these four WHO-recommended vaccines, we aimed to identify existing inequities and their associated factors. Our findings will inform policy interventions to enhance equity and improve health outcomes by strengthening the country's ability to combat vaccine-preventable diseases effectively.

## Materials and methods

### Ethics statement

This study did not collect or use any data at the individual level. The source database contained only aggregated administrative data at the institution or district/county level without any details that could identify individuals.

### Design

This is a quantitative study based on administrative data collected through a national institutional survey conducted as part of China's National Health Accounts 2022, which estimated China's health expenditures. A nationally representative sample was used in the survey. In each sampled facility, the basic facility information, health expenditure data, including vaccine prices and quantities used, were collected via a web-based reporting system.

To capture variations in vaccine use, the 2022 survey instrument specifically enhanced the vaccine-related data collection form by requesting breakdowns of data by valency (e.g., HPV2, HPV4, and HPV9) for the same vaccine type. This part of the data was used in our study for analysis.

### Sampling framework

This quantitative study was based on administrative data collected through the 2022 National Health Accounts institutional survey, which aimed to estimate China's national health expenditures. The study adopted the WHO System of Health Accounts 2011 framework and adapted it to the Chinese health system context [12].

A nationally representative, stratified multi-stage sampling design was employed. Health institutions under the National Health Commission were selected using hierarchical stratification by administrative level, including provincial, prefecture, and county levels. At the provincial level, institutions were sampled to ensure representation of major public health and medical facility types, including Centers for Disease Control and Prevention, maternal and child health institutions, health education institutions, emergency centers, blood centers, specialty public health institutions, general hospitals, traditional Chinese medicine hospitals, and specialty hospitals. If only one institution of a given type existed, it was included; if multiple institutions existed, approximately half were selected based on their information system capacity.

At the prefecture level, at least one-third of prefectures within each province were selected. Prefectures were grouped by GDP per capita, and one prefecture was selected from each group while considering geographic distribution and health service capacity. Within selected prefectures, corresponding municipal-level institutions were sampled using the same institutional coverage principles as at the provincial level.

At the county level, at least one urban district and two counties were selected within each sampled prefecture, and the total number of sampled counties accounted for no less than 15% of all counties in the province. In municipalities directly under the central government, one-quarter of districts were selected. Primary care facilities were further sampled within selected counties, including 5–8 community health centers or township health centers per county, with affiliated community health stations and village clinics selected accordingly.

In addition, the private health providers, including private hospitals, outpatient departments, and individual clinics, were sampled proportionally based on facility type to ensure comprehensive coverage of health service providers.

### Data

The data consisted of annual summary statistics covering the period from January 1 to December 31, 2022. The study sample included 2,609 public primary healthcare facilities and 355 district- and county-level centers for disease control and prevention located in both urban and rural areas across 27 out of 31 provinces in China. Each participating facility was required to complete two separate reporting forms related to vaccine provision, one for NIP vaccines and one for non-NIP vaccines.

We used two sets of vaccine data in this study. The first set of data was the number of vaccines purchased and delivered in the district and county, completed by the district- and county-level centers for disease control and prevention, which serve as intermediaries for local vaccine supply and distribution. The second set of data was at the level of vaccination clinics to represent the actual use of vaccines in the clinics. We focused on the four types of vaccines that the WHO recommends member states include in their national immunization programs for children. All vaccine products, whether single or combination formulations and regardless of valency, were included if they contained Hib, pneumococcal, rotavirus, or HPV components.

We obtained the vaccine purchase data from 370 district- and county-level centers for disease control and prevention in China. A total of 15 centers did not complete the vaccine data form and were excluded from the analysis. Among the excluded centers, seven were located in counties, seven in urban districts, and one in a county-level city, and they were distributed across 12 provinces. No significant differences were observed between excluded and included centers in terms of geographic distribution, administrative level, or key institutional characteristics. A final sample of 355 districts and counties in 27 Chinese provinces were included for analysis.

In addition, among 5247 health facilities, we excluded 2423 health facilities that were not public primary health care facilities. A total of 2824 public primary health facilities were identified, among which 124 reported no vaccination record and were therefore excluded from the analysis. Another 77 facilities were also excluded due to an unusually high price for vaccines, and it could not be determined if they misentered the price or volume. Also, 14 facilities that did not provide NIP vaccines were excluded, as they were not primary providers of services to children, the main target population for the vaccines of interest in this study. As a result, 2609 public primary health facilities were included for the final analysis. These facilities were located in 410 districts and counties in 27 provinces in China.

Supplementary data on socio-economic indicators, such as provincial GDP per capita in 2022, were sourced from publicly available database provided by the National Statistics Bureau [13]. These indicators were used to assess equity in vaccine availability by comparing them across provinces.

## Vaccines

Based on the survey data, we identified a total of ten vaccine products that fall within the four vaccine types recommended by the World Health Organization (WHO) for inclusion in national immunization programs. These include four Hib-containing vaccines: 1) Haemophilus influenzae type b conjugate vaccine (Hib); 2) Meningococcal groups A and C and Haemophilus influenzae type b conjugate vaccine (MenAC-Hib); 3) Diphtheria, tetanus, acellular pertussis, and Haemophilus influenzae type b combined vaccine (DTaP-Hib); and 4) Diphtheria, tetanus, pertussis (acellular, component), poliomyelitis (inactivated), and Haemophilus influenzae type b conjugate vaccine, adsorbed (DTaP-IPV-Hib). The pneumococcal conjugate vaccine (PCV) category was represented by a single product: the 13-valent pneumococcal polysaccharide conjugate vaccine (PCV13). The rotavirus vaccine category included two products: 1) Rotavirus (live) vaccine, oral, commonly known as the Lanzhou lamb rotavirus vaccine (LLR); and 2) Reassortant rotavirus vaccine, live, oral, pentavalent (vero cell) (RV5). The human papillomavirus (HPV) vaccine category included three formulations based on valency: 1) Bivalent HPV vaccine (types 16 and 18), adsorbed; 2) Quadrivalent recombinant HPV vaccine (types 6, 11, 16, and 18); and 3) 9-valent recombinant HPV vaccine (types 6, 11, 16, 18, 31, 33, 45, 52, and 58). During the study period in 2022, HPV vaccines were licensed only for females in China; therefore, all HPV vaccine availability measures in this study pertain to female vaccination services.

## Analysis

We first conducted a descriptive analysis to examine disparities in vaccine availability across China's economic regions and residential settings. We used the classification by the National Bureau of Statistics, provinces were grouped into eastern, central, and western regions, which has been widely used in socioeconomic and policy research in China. Residential

settings were defined using administrative classifications, with districts and county-level cities classified as urban, and counties as rural, reflecting typical patterns of economic activity. At the clinic level, community health service centers were considered urban, and township hospitals were considered rural. This classification enabled a consistent comparison of vaccine availability across geographic and socioeconomic contexts.

Vaccine availability is defined as the presence of a specific vaccine at a given location of immunization service delivery and time period. In this study, it was measured using a binary variable: at the district and county level, 1 if at least one dose of a specific vaccine was purchased or distributed by centers for disease control and prevention, and 0 otherwise; at the clinic level, 1 if at least one dose of a specific vaccine was administered, or 0 otherwise. The centers for disease control and prevention at district and county level collect the purchasing numbers from each vaccination clinic and therefore its purchasing amounts represent an overall availability in the district and county. We examined vaccine availability along two dimensions: a) the number and percentage of districts or counties and clinics where a certain type of vaccine was available, and b) the number of the four types of vaccines of interest in this study, specifically, whether 1, 2, 3, or all 4 types were available.

When analyzing vaccine availability, we began by examining the aggregated availability of different vaccine types. Following the WHO's position papers and recommended immunization schedules, some vaccines may appear in single-antigen form or as components of combination vaccines. In such cases, we group vaccines according to the WHO's recommendations. For example, any vaccine containing Hib is classified under the Hib vaccine group. This approach allows for a more explicit measurement of vaccine availability.

To reflect the extent of absolute and relative inequality in the availability of vaccines of interest, we used the slope index of inequality (SII) and the concentration index (CI). These two indicators were calculated based on the ranked socioeconomic status of provinces, allowing us to capture both the magnitude and direction of inequality across the distribution.

The SII measures absolute inequality by estimating the difference in vaccine availability across provinces, taking into account the entire distribution of socioeconomic status represented by provincial GDP per capita. A positive SII indicates that vaccine availability is higher in provinces with higher socioeconomic status (pro-rich), while a negative SII suggests higher availability in poorer provinces (pro-poor). An SII value close to zero indicates minimal absolute inequality.

The CI, on the other hand, measures relative inequality by assessing the extent to which vaccine availability is concentrated among populations with higher or lower socioeconomic status, in this study, provinces with higher or lower GDP per capita. A positive CI means availability is disproportionately concentrated in wealthier provinces, while a negative CI suggests the opposite. A CI value near zero indicates that vaccine availability is relatively evenly distributed, whereas values further from zero indicate greater inequality. We used the Erreygers-corrected CI to measure relative inequalities due to the binary nature of the outcome variable [14].

In addition, we plotted concentration curves to illustrate the cumulative distribution of vaccine availability across the socioeconomic gradient as a comprehensive view of how inequality varied across types of vaccines.

All calculations were conducted using Stata 18 (StataCorp LLC, College Station, TX, USA).

## Results

### Vaccine availability at district- and county-level

We examined the availability of four vaccine types, namely, Hib, PCV, rotavirus, and HPV, across 355 districts and counties in China. Overall, all four vaccines were available in over 85% of districts and counties, with HPV being the most widely available (96.9%), and PCV the least (86.5%). By economic region, the eastern region showed the highest availability for all vaccine types, while the western region had the lowest availability for Hib, pneumococcal, and rotavirus vaccines, and the central region had the lowest HPV vaccine availability at 94.2%. When comparing residential types, vaccine availability was consistently higher in urban than in rural areas. The greatest disparity was observed in PCV availability, which was 94.2% in urban areas compared with 79.1% in rural areas. The availability of vaccines was presented in Table 1.

**Table 1. Vaccine stock availability at the district/county level by region and between urban and rural areas in China, 2022 (n, %).**

| | Number of districts/counties providing 4 vaccines | | | | Total number of districts/counties |
| --- | --- | --- | --- | --- | --- |
| | Hib | Pneumococcal | Rotavirus | HPV | |
| **Total** | 321 (90.4%) | 307 (86.5%) | 321 (90.4%) | 344 (96.9%) | 355 |
| **Economic region** | | | | | |
| Eastern | 119 (97.5%) | 115 (94.3%) | 115 (94.3%) | 121 (99.2%) | 122 |
| Central | 94 (90.4%) | 90 (86.5%) | 94 (90.4%) | 98 (94.2%) | 104 |
| Western | 108 (83.7%) | 102 (79.1%) | 112 (86.8%) | 125 (96.9%) | 129 |
| **Residential type** | | | | | |
| Urban | 166 (96.0%) | 163 (94.2%) | 165 (95.4%) | 170 (98.3%) | 173 |
| Rural | 155 (85.2%) | 144 (79.1%) | 156 (85.7%) | 174 (95.6%) | 182 |

This table presents the number and proportion of sampled districts and counties that offer specific types of vaccines. Hib includes Hib, MenAC-Hib, DTaP-Hib, and DTaP-IPV-Hib. Pneumococcal refers to PCV13. Rotavirus includes LLR and RV5. HPV includes HPV2, HPV4, and HPV9.

Vaccine availability varied substantially across economic regions and between urban and rural settings. Over 90% of districts and counties in the eastern region and the urban areas provided all four types of WHO-recommended vaccines in 2022. However, in the central and western regions, we observed a lower availability of these vaccines, and the rural areas showed a similar pattern to that in western region where only 75% of districts and counties could provide all four types of vaccines. Fig 1 illustrates the distribution of districts and counties by the number of vaccine types available.

## Equity in vaccine availability at the point of immunization service delivery

At clinic level, we found that the availability of specific vaccines varied across economic regions and residential areas as well. The eastern region showed the highest availability (50.7%-73.1%) for most vaccines, except for MenAC-Hib (0.7%) and DTaP-Hib (28.2%). The central region showed a moderate level (26.0%-60.5%, excluding MenAC-Hib) of availability, while the western region had the lowest availability (16.2%-48.3%, excluding MenAC-Hib) for most vaccines. The disparities in availability between urban and rural areas are significant, with most vaccines available at around 80–90% in urban areas but below 50% in rural areas.

Vaccine availability also differed when comparing specific vaccines. For example, the availability of DTaP-IPV-Hib was higher than DTaP-Hib in the eastern region and urban areas, whereas the opposite was true in the central and western

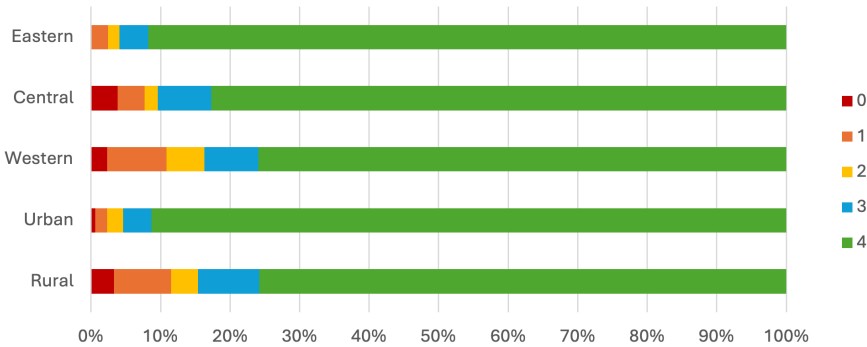

**Fig 1. Distribution of districts/counties by the number of vaccines available, 2022.**

regions and rural areas. Similarly, the availability of RV5 was higher in the eastern region and urban areas, but lower in the central and western regions and rural areas. The availability of each vaccine was shown in Table 2.

By analyzing the distribution based on the number of WHO-recommended vaccine types, we observed a pattern similar to that of vaccine availability. A higher proportion of vaccination clinics in the eastern region and urban areas offered all four types of vaccines. This proportion decreased in the central and western regions, as well as in rural areas. More than 50% of clinics in the eastern region and over 70% in urban areas offered all four vaccine types, whereas only about 20% in the western region and 40% in rural areas did so. Meanwhile, more than 30% of clinics in the western region and 20% in rural areas did not provide any of these vaccines. The distribution of these clinics is shown in Fig 2.

We found clear evidence of both absolute and relative inequality in the availability of four types of vaccines recommended by WHO-recommend in China. The slope index of inequality (SII) was statistically significant for LLR (1.906), HPV2 (1.950), HPV9 (2.504), RV5 (2.508), HPV4 (2.924), PCV13 (2.964), and DTaP-IPV-Hib (3.648), indicating an increasingly unequal distribution pattern. Similarly, the Concentration index (CI) was statistically significant for DTaP-Hib (0.147), Hib (0.209), HPV2 (0.213), LLR (0.230), HPV9 (0.311), HPV4 (0.314), RV5 (0.343), PCV13 (0.362), and

**Table 2. Availability of vaccines at clinics, by location, 2022 (%).**

| Vaccines | Eastern | Central | Western | Urban | Rural | Total |
|---|---|---|---|---|---|---|
| **Hib-containing** | 78.8 | 67.6 | 48.6 | 89.6 | 58.4 | 65.3 |
| Hib | 69.0 | 60.5 | 45.5 | 76.9 | 53.3 | 58.5 |
| MenAC-Hib | 0.7 | 2.1 | 1.2 | 1.7 | 1.1 | 1.3 |
| DTaP-Hib | 28.2 | 30.6 | 19.0 | 55.8 | 17.1 | 25.7 |
| DTaP-IPV-Hib | 50.7 | 26.0 | 16.2 | 75.5 | 19.6 | 32.0 |
| **PCV13** | 62.9 | 50.8 | 30.1 | 87.9 | 36.9 | 48.3 |
| **Rotavirus-containing** | 69.4 | 62.4 | 44.3 | 89.6 | 50.0 | 58.8 |
| LLR | 55.9 | 52.1 | 37.7 | 79.4 | 39.8 | 48.6 |
| RV5 | 57.8 | 44.9 | 28.1 | 78.9 | 34.0 | 44.0 |
| **HPV-containing** | 82.8 | 67.1 | 58.0 | 90.7 | 63.9 | 69.9 |
| HPV2 | 69.9 | 58.5 | 48.3 | 81.7 | 52.9 | 59.3 |
| HPV4 | 73.1 | 45.6 | 42.7 | 78.6 | 48.3 | 55.0 |
| HPV9 | 67.1 | 45.7 | 37.5 | 74.4 | 44.3 | 51.0 |

Note: Table 1 reports availability at the district or county level, defined as the presence of at least one clinic providing the vaccine. Table 2 reports availability at the clinic level. Because not all clinics within a district or county necessarily offer the vaccine, clinic-level percentages are lower and the two tables reflect different units of analysis.

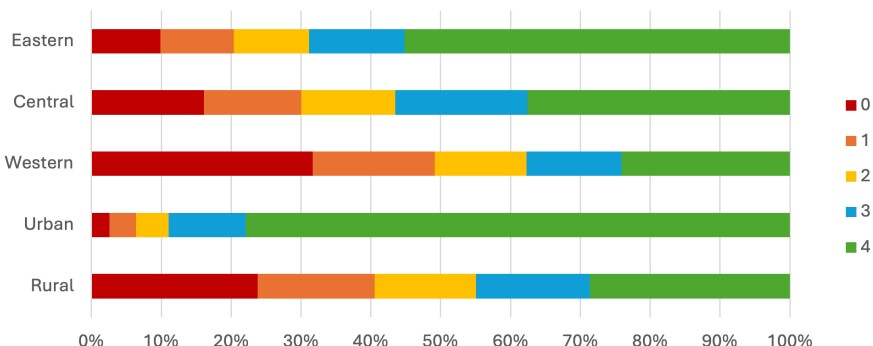

**Fig 2. Distribution of vaccination clinics by the number of vaccine types provided, 2022.**

DTaP-IPV-Hib (0.408). Both measures of inequality consistently indicated that DTaP-IPV-Hib, PCV13, RV5, HPV4, and HPV9 were more commonly available in economically advantaged provinces, reflecting higher levels of inequality. Detailed results can be found in Table B in S1 Appendix.

Fig 3 shows that vaccine availability was generally concentrated in regions with higher economic status, as indicated by the concentration curves. Among all vaccines, DTaP-IPV-Hib exhibited the greatest inequality, with a concentration curve

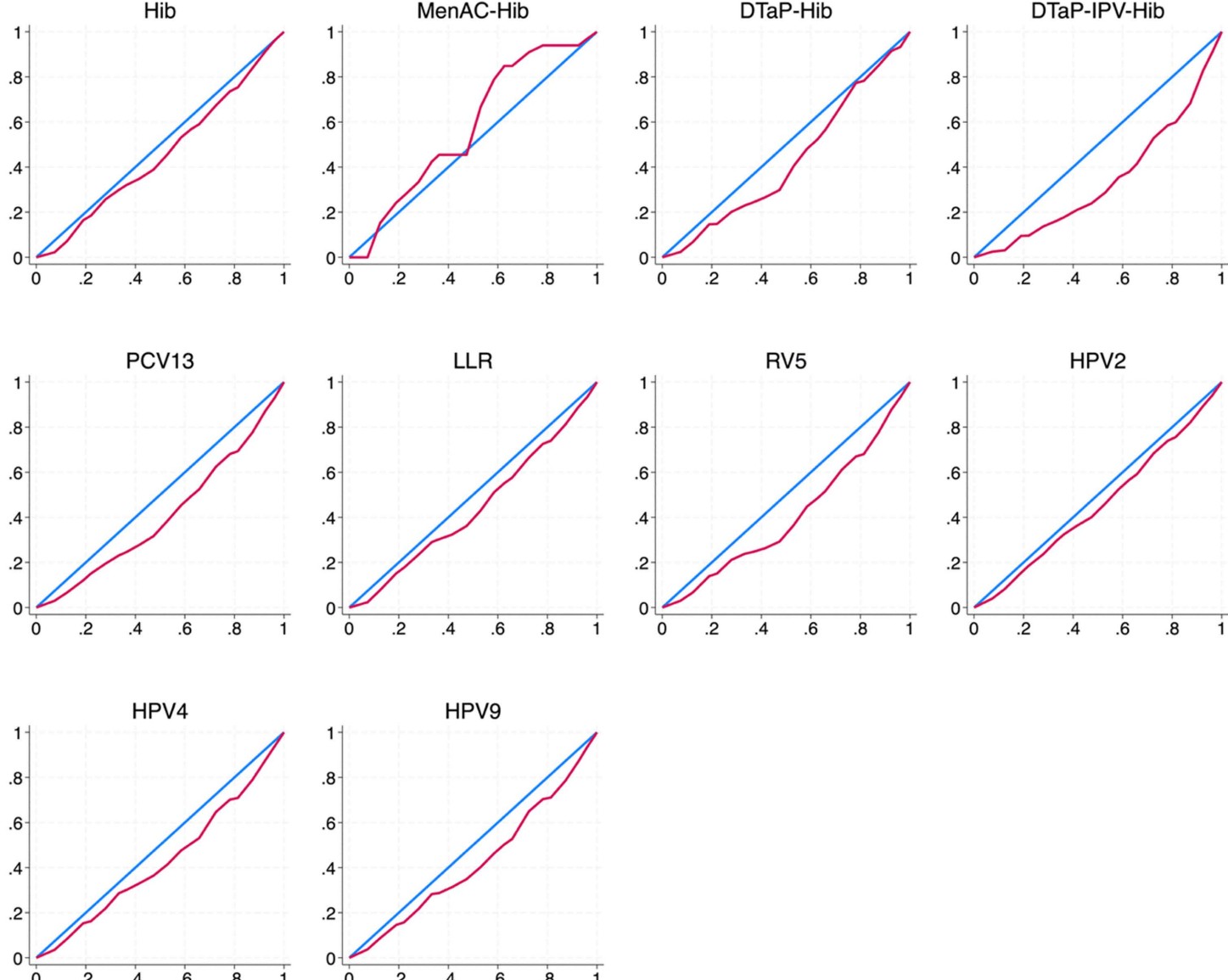

**Fig 3. Concentration curve on vaccine availability, 2022.** Note: This figure shows the concentration curve of vaccine availability. The x-axis represents the cumulative percentage of clinics, ranked by provincial GDP per capita from the lowest to the highest. The y-axis represents the cumulative percentage of clinics providing a specific type of vaccine. If the curve lies above the line of equality (the 45-degree line), it indicates greater availability in economically disadvantaged areas. If the curve lies below the line of equality, it indicates greater availability in economically better-off areas.

well below the line of equality, followed by PCV13, RV5, HPV9, and others. Hib and HPV2 also fell below the equity line, but demonstrated relatively less inequality. In contrast, MenAC-Hib was more concentrated in economically disadvantaged provinces. These patterns are consistent with the corresponding CI results.

## Discussion

### Summary of key findings

This study is, as far as we are aware of, the first study to analyze the vaccine availability from provider side in China. It examined three equity issues in vaccine availability across China: regional disparities, urban–rural divides, and differences among the WHO-recommended 4 vaccines, namely, Hib, pneumococcal, rotavirus, and HPV vaccines. Our analysis indicates that the economically advanced eastern regions have higher availability for most vaccines, whereas the central and western regions show comparatively lower availability. The disparity is larger between urban and rural areas, with urban centers generally having better access to vaccines than rural areas. Furthermore, certain vaccines, particularly newer and higher-valent ones such as the 9-valent HPV vaccine and PCV13, demonstrate greater levels of inequality in availability. Among all the vaccines examined in this study, DTaP-IPV-Hib displayed the highest level of inequality in availability.

### Disparities in vaccine availability

We found significant regional disparities in vaccine availability, reflecting the inequal provision of vaccine among China's economic regions at both districts/county and clinic level. Our findings echo other analysis on the coverages of these vaccines, such as DTaP-IPV-Hib which the coverage rate in eastern provinces were higher than other regions [15]. Such regional disparities were commonly seen in countries where the economic development was geographically uneven as similar patterns was observed in the Unites States [16] and India [17].

Another evident disparity exists between urban and rural areas. In our findings, the availability of these non-NIP vaccines was much lower in rural areas, which was in accordance to the lower vaccination rate in rural aeras [18]. In another study, urban-rural residence account for 27.2% of inequality for the third dose of PCV vaccine in China, indicating the role of vaccine provision in the equity issue [11]. The difference of vaccination rate between urban and rural areas were common in the world mostly [19,20].

We also found the internal disparities in the availability of these four types of vaccines. We found that the equity was lowest for PCV13 (about USD 87.54/dose, USD 350.14/full course, USD 1 = CNY 7.1) and the DTaP-IPV-Hib vaccine (about USD 85.63/dose, USD 342.54/full course), which may be related to their relatively high prices and low awareness. Interestingly, in the central region, the coverage rate of the quadrivalent vaccine (about USD 53.10/dose, USD 212.39/full course) is higher than that of the DTaP-IPV-Hib vaccine, whereas in the eastern region, the DTaP-IPV-Hib vaccine coverage exceeds that of the quadrivalent vaccine, reflecting regional differences in vaccine preferences corresponding to economic development levels. A similar pattern is observed with rotavirus vaccines: the imported pentavalent rotavirus vaccine (about USD 40.63/dose, USD 121.89/full course) is more popular in the eastern region, compared to the monovalent domestic product (about USD 25.25/dose, USD 75.74/full course). Among these vaccines, the Hib conjugate vaccine has the lowest price (about USD 14.79/dose, USD 44.37/full course) and also the highest equity in access. This demonstrates a certain relationship between vaccine price and equity. Consequently, the vaccination rate of these vaccines were affected [15,21] accordingly.

### Main factors influencing vaccine availability

The decision to make certain types of non-NIP vaccines available in the district/county is rooted in the procurement process of these non-NIP vaccines. In China, unlike the universal access to NIP vaccines, the provision of non-NIP vaccines varies across provinces due to differences in procurement processes, manufacturer availability, and local demand. Each district- and county- level center for disease control and prevention is responsible for estimating the demand from

vaccination clinics within its jurisdiction. These estimates are then consolidated, and orders are placed through the province's public resource procurement platform [22]. However, not all vaccines are available for procurement on every provincial platform, nor are all vaccines procured by every vaccination clinic. This selective availability is jointly influenced by manufacturers, vaccination clinics, and vaccine recipients.

Vaccine manufacturers were facing challenges when supplying vaccines to remote areas. As China's vaccine procurement and distribution system was tightly regulated, manufacturers were often less willing to take on distribution responsibilities in areas with low population density or difficult geographic conditions, such as remote or mountainous regions, where delivery costs were high [22].

Another important dimension is the variation in service capacity. A study shows the eastern regions have higher service capacity in terms of human resources and equipment [23]. Meanwhile, the opening days were significantly lower in western regions, as 20.9% of vaccination clinics open on a monthly basis (one day in a month) while only 3.4% in eastern regions. The service capacity in rural areas was also constrained in provision of these vaccines, as they were suffering from low retention of healthcare workers, shortage of professionals, and insufficient financial incentives [24]. Limited service capacity might reduce the willingness to provide additional services other than the NIP vaccination, which was required by law. Before 2016, price of vaccine contained markup to cover actual service cost and serve as an incentive to clinics; after the policy reform, the vaccine price markup was canceled and a fixed price service fee were used to cover the service cost. However, the service fee for non-NIP vaccines was about CNY 20–25 (USD 2.82 – 3.52, USD 1 = CNY 7.1) per dose, which was not enough to cover the cost in some areas.

From the demand-side perspective, caregivers' financial capacity, awareness, and willingness to pay are key factors influencing vaccine uptake. These factors are often closely linked to residents' socioeconomic status and the level of local economic development, both of which ultimately affect affordability when the price of some vaccines is high. The pricing strategy of these non-NIP vaccines are different and complex in China's market [5]. When no competition exist for certain type of vaccines, such as PCV13 and DTaP-IPV-Hib in 2022, their price may reach as high as that for the private sector in the United States, and about 30 times the price for UNICEF [25]. One key factor contributing to the regional disparities is the income level when residents in more developed areas tend to have higher incomes than those in central and western regions, enabling them to afford higher-priced vaccines from an out-of-pocket cost perspective. Analysis found that the family wealth contributed 40.9% of the inequality in the third dose of PCV, which is full out-of-pocket payments, while only 2.7% when examining the equity for full-course NIP vaccines, which are free [11]. Some heads of immunization services also indicated that caregivers' willingness to vaccinate their children is a key factor influencing whether local clinics procure certain types of non-NIP vaccines [24].

Lastly, policy and regulation are important influential factors shaping vaccine availability in China. Unlike many other countries, China mandates the use of certain vaccines for children through national legislation, making them compulsory. Proof of vaccination is required for enrollment in preschool and elementary education, which further reinforces uptake. The availability of these required vaccines, those included in the NIP, is largely ensured throughout the supply chain, from procurement to distribution [5]. Given the predictable demand based on birth cohorts and centralized procurement, manufacturers face lower production risks and can plan production more efficiently. At the point of service delivery, Chinese law requires each district and county to designate at least one vaccination clinic to provide NIP vaccines, ensuring broad geographic access. As a result, the availability of NIP vaccines is high, which is reflected in coverage data consistently exceeding 90% in stark contrast to the coverage levels of the four major non-NIP vaccines in our study [11].

### Limitations

This study has several limitations. While it aims to assess vaccine availability, data constraints necessitated the use of actual provision data as a proxy. Vaccine availability was operationalized in a binary manner, defined as present if at least one dose was administered during the study year and absent otherwise. Consequently, instances where a vaccine was

listed on the menu but not administered throughout the study year were not captured. In addition, this binary classification does not capture temporal variations in supply within the year. It may underestimate chronic shortages or intermittent stock outs. Clinics experiencing prolonged supply constraints over an extended period may still have been classified as having availability if any doses were administered during the year. Similarly, short term stock interruptions that temporarily restricted access would not be reflected in the annual binary measure. These limitations may lead to imprecision in estimating the stability and continuity of vaccine supply.

As a cross-sectional study focusing on a single year, it cannot account for clinics that commenced operations mid-year or later, resulting in incomplete annual data. Additionally, the findings may be influenced by significant events during the study period, such as pandemic-related emergency vaccination campaigns, which could have temporarily affected vaccine provision patterns. Rapid policy changes further complicate the study's relevance, as the data may not reflect current conditions. Several provinces have recently integrated the HPV vaccine into their local immunization programs, likely enhancing its availability beyond what is represented in this study. In addition, since the completion of this study, the bivalent HPV vaccine has been incorporated into China's National Immunization Program for 13-year-old females, which is expected to substantially expand its nationwide availability [26]. Therefore, the findings presented here should be interpreted within the policy context of 2022 and may not fully reflect subsequent national-level expansions.

## Conclusion

This study provides the first comprehensive analysis of non-NIP vaccine availability from the provider side in China, uncovering substantial inequities across regions, urban and rural areas, and vaccine types. These disparities are driven by a constellation of factors across vaccine supply and service demand. To address these inequities, policy interventions should consider expanding the scope of publicly funded vaccination and new vaccine introduction into NIP, harmonizing procurement mechanisms across provinces, and strengthening the service capacity of under-resourced areas. As China moves toward building a more equitable and life-course immunization system, ensuring fair access to high-quality vaccines for all populations, regardless of geography or socioeconomic status, should remain a core priority.

## Supporting information

**S1 Appendix. Table A.** Vaccine availability at clinic level, 2022 (n, %). **Table B.** Absolute and relative inequality on vaccine availability, 2022. **Table C.** Classification of economic regions in China. **Table D.** Data collection table - free vaccines included in the National Immunization Program (NIP), 2022. **Table E.** Data collection table - self-paid vaccines (non-NIP vaccines), 2022.
(DOCX)

## Acknowledgments

We thank the county- and district-level Centers for Disease Control and Prevention and public primary healthcare facilities across China that participated in the 2022 National Health Accounts institutional survey and contributed the data used in this study. We also acknowledge the teams involved in coordinating and implementing the survey across the country.

## Author contributions

**Conceptualization:** Lei Guo, Shenglan Tang.

**Data curation:** Lei Guo, Feng Guo, Weixi Jiang, Xinyu Zhang, Shu Chen, Quan Wan.

**Formal analysis:** Lei Guo.

**Funding acquisition:** Shenglan Tang.

**Investigation:** Lei Guo.

**Methodology:** Lei Guo.

**Visualization:** Lei Guo.

**Writing – original draft:** Lei Guo.

**Writing – review & editing:** Feng Guo, Weixi Jiang, Xinyu Zhang, Di Dong, Shu Chen, Quan Wan, Shenglan Tang.

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
