## [Decision Letter · Decision Letter 0]

22 Feb 2026

PGPH-D-25-02255

Availability of Hib, Pneumococcal, Rotavirus, and HPV vaccines in China: Implication for equity in access to immunization services

Dear Dr. Guo,

Thank you for submitting your manuscript to PLOS Global Public Health. After careful consideration, we feel that it has merit but does not fully meet PLOS Global Public Health’s publication criteria as it currently stands. Therefore, we invite you to submit a revised version of the manuscript that addresses the points raised during the review process.

EDITOR:

Line 101-105: Add details regarding sampling framework.

Line 125-127: Clarify if 15 centres that did not complete the vaccine data form differed from the 355 districts that were included for analysis.

Line 128-135: Clarify if 2,824 public primary health facilities that had data extracted differed from 2,423 who did not. Also how other exclusions may have affected results.

Line 136-138: Add references for databases

Line 208 Results: Is there any data for vaccine coverage (i.e. how many children received vaccines) in each of the regions.

Line 517-518: Specify the wealth status of the three economic regions.

Correct minor spelling and grammatical errors.

We look forward to receiving your revised manuscript.

Kind regards,

Claire E. von Mollendorf

Academic Editor

Journal Requirements:

i. Please clarify all sources of financial support for your study. List the grants, grant numbers, and organizations that funded your study, including funding received from your institution. Please note that suppliers of material support, including research materials, should be recognized in the Acknowledgements section rather than in the Financial Disclosure.

ii. State the initials, alongside each funding source, of each author to receive each grant. For example: "This work was supported by the National Institutes of Health (####### to AM; ###### to CJ) and the National Science Foundation (###### to AM)."

iii. State what role the funders took in the study. If the funders had no role in your study, please state: “The funders had no role in study design, data collection and analysis, decision to publish, or preparation of the manuscript.”

iv. If any authors received a salary from any of your funders, please state which authors and which funders.

2. Please ensure that your Ethics Statement is available in its entirety at the beginning of your Methods section, under a subheading 'Ethics Statement'.

3. Please upload separate figure files in .tif or .eps format. Also, remove the figures from your manuscript file but keep the legends.

4. We notice that your supplementary tables are included in the manuscript file. Please remove them and upload them with the file type 'Supporting Information'. Please ensure that each Supporting Information file has a legend listed in the manuscript after the references list.

5. We note that you have indicated that there are restrictions to data sharing for this study. PLOS only allows data to be available upon request if there are legal or ethical restrictions on sharing data publicly. For more information on unacceptable data access restrictions, please see http://journals.plos.org/plosone/s/data-availability#loc-unacceptable-data-access-restrictions.

Additional Editor Comments (if provided):

Reviewers' comments:

Reviewer's Responses to Questions

**Comments to the Author**

1. Does this manuscript meet PLOS Global Public Health’s publication criteria? Is the manuscript technically sound, and do the data support the conclusions? The manuscript must describe methodologically and ethically rigorous research with conclusions that are appropriately drawn based on the data presented.? Is the manuscript technically sound, and do the data support the conclusions? The manuscript must describe methodologically and ethically rigorous research with conclusions that are appropriately drawn based on the data presented.

Reviewer #1: Yes

Reviewer #2: Yes

2. Has the statistical analysis been performed appropriately and rigorously?

Reviewer #1: Yes

Reviewer #2: Yes

3. Have the authors made all data underlying the findings in their manuscript fully available (please refer to the Data Availability Statement at the start of the manuscript PDF file)?

The PLOS Data policy requires authors to make all data underlying the findings described in their manuscript fully available without restriction, with rare exception. The data should be provided as part of the manuscript or its supporting information, or deposited to a public repository. For example, in addition to summary statistics, the data points behind means, medians and variance measures should be available. If there are restrictions on publicly sharing data—e.g. participant privacy or use of data from a third party—those must be specified.requires authors to make all data underlying the findings described in their manuscript fully available without restriction, with rare exception. The data should be provided as part of the manuscript or its supporting information, or deposited to a public repository. For example, in addition to summary statistics, the data points behind means, medians and variance measures should be available. If there are restrictions on publicly sharing data—e.g. participant privacy or use of data from a third party—those must be specified.

Reviewer #1: Yes

Reviewer #2: No

4. Is the manuscript presented in an intelligible fashion and written in standard English?

Reviewer #1: Yes

Reviewer #2: Yes

Reviewer #1: This is a very interesting paper but a few pieces of additional information would make it stronger they include:

1. Foregrounding the prices earlier in the paper AND stating whether the clinic or district needs to outlay money to purchase a stock. To do so, I would add a table of the prices of the vaccines earlier on, and in that table i would clarify 1. Hib (a,b,c,d types), 2 PCV (a.b.c etc.) so that when the reader is looking at the data it is clear how much financial burden is not only on the family but on the clinics themselves. Because you state the the demand side analysis has been done, it should be cited more clearly. Similarly, i would add the give the population # and eligible population # for each region (children under 5, girls 11 etc.) as well, if possible as the average household or per capita income of the regions in question.

2.If possible, document the differential health budgets for each region or district (per capita if possible)

3. Explaining more clearly the difference between Table 1--which shows rates of 80s-90s % for Hib, and Table 2 which shows 50-70% These two tables seem to contradict one another.

4. In the study limitations, please include the limits of description of vaccine in the binary fashion you chose (present (even one dose) or absent) this likely under estimates chronic shortages and intermittent stock outs which can drive down demand.

I

Reviewer #2: The investigators report an ERC-exempt, Gates Foundation-funded study of the availability at the clinic and county level of four non-program vaccines in China that WHO recommends that all national immunization programs include – HPV, PCV, Hib, and rotavirus vaccines. The study was based on analysis of 2022 survey (the China National Health Accounts Study) conducted by the China National Health Development Research Center, a non-governmental technical center affiliated with the National Health Commission (China’s MoH) that provides data for policy makers. The study purpose is “… to investigate regional and urban-rural disparities in vaccine availability across China.” They found that there were significant disparities in the availability of the four non-program vaccines at the clinic level and at the county level, with availability relate to socioeconomic development level. Availability was positively associated with wealth of the jurisdiction and was greater in urban compared with rural settings. They concluded that their findings highlight “significant inequities in the availability of WHO-recommended non-NIP vaccines across China, particularly affecting rural and economically disadvantaged regions,” and that “the highest inequalities were observed for higher-priced vaccines such as DTaP-IPV-Hib and PCV13.” They recommend that “… appropriate policy should be developed to prioritize expanding vaccine provision in underserved areas, strengthening rural service capacity, and considering the inclusion of these vaccines in the NIP (the National Immunization Program).”

Vaccines only work when used, and availability is essential for using a vaccine. As the authors state, the four study vaccines are recommended for all countries. Thus, the topic is important. The methods for assessing availability were appropriate, within the constraints of their data source. The figures and tables are excellent. Their conclusions were supported by the data presented, and their recommendations also follow from the data presented.

I have only a few suggestions to improve this very good manuscript.

Readers will appreciate more information about the actual survey that they used in their analysis. The reader will benefit from knowing what the China National Health Development Research Center is (e.g., government, independent, or government-affiliated), and the survey methods, ideally including response rate (if applicable). Currently, the manuscript’s description of the methods is made by referral to reference 13, which is an independent analysis published in 2020 of an “adolescent health expenditure in this study covered the period from 1st January to 31st December 2014,” quoting reference 13. This survey was in 9 provinces, but the present (2022) survey was in 27 provinces. Perhaps the authors could provide a brief paragraph on the present survey’s sampling strategy and sampling results. The reader would also benefit from being able to view the survey instrument (which had been augmented for vaccine access) in supplementary materials (or a link to the survey instrument in the references).

Lines 73-75 – The sentence about program vaccines states that the vaccines that are included in the National Immunization Program “… are provided free of charge to children under six years old.” The upper age limit of the program was extended to 18 years of age before 2021. For example, the schedule states that “children below 18 years of age who have not completed their NIP vaccination series by the recommended ages should be caught up based on the following principles …” (National Health Commission of The People's Republic of China. Childhood Immunization Schedule for National Immunization Program Vaccines - China (Version 2021). China CDC Wkly. 2021 Dec 24;3(52):1101-1108. doi: 10.46234/ccdcw2021.270). The authors could indicate that during the study period (2022), there were no program vaccines routinely recommended beyond 6 years of age, but catch-up vaccination is free of charge through 18 years.

During the study period, there were no HPV vaccines licensed for males in China. The authors should indicate that all HPV vaccine availability measures in the present study pertained to females only.

In the limitations section, the authors mention that “Rapid policy changes further complicate the study’s relevance, as the data may not reflect current conditions. Several provinces have recently integrated the HPV vaccine into their local immunization programs, likely enhancing its availability beyond what is represented in this study.” The authors should indicate that HPV2 vaccine is now included in the National Immunization Program for 13-year-old females (Wang F, Cao L, Li K, et al. Technical and Implementation Guidelines for the Introduction of Human Papillomavirus Vaccine into China's National Immunization Program. China CDC Wkly. 2025 Dec 12;7(50):1545-1548. doi: 10.46234/ccdcw2025.262).

I was unable to follow the link in reference 12, receiving a 404 response. Authors should check to verify access to that website.

**Do you want your identity to be public for this peer review?** For information about this choice, including consent withdrawal, please see our Privacy Policy..

Reviewer #1: **Yes:** joia mukherjeejoia mukherjeejoia mukherjeejoia mukherjee

Reviewer #2: No

---

## [Decision Letter · Decision Letter 1]

8 Apr 2026

Availability of Hib, Pneumococcal, Rotavirus, and HPV vaccines in China: Implication for equity in access to immunization services

PGPH-D-25-02255R1

Dear Mr Guo,

We are pleased to inform you that your manuscript 'Availability of Hib, Pneumococcal, Rotavirus, and HPV vaccines in China: Implication for equity in access to immunization services' has been provisionally accepted for publication in PLOS Global Public Health.

Best regards,

Claire E. von Mollendorf

Academic Editor

Reviewer Comments (if any, and for reference):

Reviewer's Responses to Questions

**Comments to the Author**

Reviewer #2: All comments have been addressed

publication criteria? Is the manuscript technically sound, and do the data support the conclusions? The manuscript must describe methodologically and ethically rigorous research with conclusions that are appropriately drawn based on the data presented.? Is the manuscript technically sound, and do the data support the conclusions? The manuscript must describe methodologically and ethically rigorous research with conclusions that are appropriately drawn based on the data presented.

Reviewer #2: Yes

3. Has the statistical analysis been performed appropriately and rigorously?

Reviewer #2: Yes

4. Have the authors made all data underlying the findings in their manuscript fully available (please refer to the Data Availability Statement at the start of the manuscript PDF file)?

The PLOS Data policy requires authors to make all data underlying the findings described in their manuscript fully available without restriction, with rare exception. The data should be provided as part of the manuscript or its supporting information, or deposited to a public repository. For example, in addition to summary statistics, the data points behind means, medians and variance measures should be available. If there are restrictions on publicly sharing data—e.g. participant privacy or use of data from a third party—those must be specified.requires authors to make all data underlying the findings described in their manuscript fully available without restriction, with rare exception. The data should be provided as part of the manuscript or its supporting information, or deposited to a public repository. For example, in addition to summary statistics, the data points behind means, medians and variance measures should be available. If there are restrictions on publicly sharing data—e.g. participant privacy or use of data from a third party—those must be specified.

Reviewer #2: Yes

5. Is the manuscript presented in an intelligible fashion and written in standard English?

Reviewer #2: Yes

Reviewer #2: I was one of the reviewers of the original manuscript. The authors have addressed my comments and suggestions in their response letter and in the manuscript, and I have no additional comments or suggestions.

**Do you want your identity to be public for this peer review?** For information about this choice, including consent withdrawal, please see our Privacy Policy..

Reviewer #2: No
